# The Use of Fish Scale Hormone Concentrations in the Assessment of Long-Term Stress and Associated Adverse Effects on Reproductive Endocrinology

**Emily K. C. Kennedy** [1,*,†] **and David M. Janz** [2]

1    Toxicology Graduate Program, University of Saskatchewan, Saskatoon, SK S7N 5B3, Canada
2    Western College of Veterinary Medicine and Toxicology Centre, Saskatoon, SK S7N 5B3, Canada
*    Correspondence: emily.kennedy@usask.ca; Tel.: 1-306-966-4147
†    This work was part of the Master of Science thesis of Emily Kathleen Colette Kennedy.
     Master of Science program at the University of Saskatchewan, Saskatoon, SK, Canada.

**Abstract:** Investigation of the use of fish scales as a medium for non-lethal biomonitoring has recently commenced. Fish scales have been shown to incorporate cortisol over longer periods of time than blood and thus provide a promising means of assessing long-term stress in many species of teleost fish. However, while cortisol is a major mediator of the stress response in fishes, downstream effects of chronic stress on reproduction can involve gonadal steroids such as progesterone and testosterone. The quantification of these additional hormones alongside cortisol could therefore allow for the assessment of both stress and consequential reproductive alterations. To investigate these concepts, we artificially elevated circulating cortisol, progesterone, and testosterone in rainbow trout (*Oncorhynchus mykiss*) using coconut oil implants for three weeks. Following this we quantified these three hormones as well as 11-ketotestosterone, a potent androgen in teleost fishes. In all cases serum samples reflected a significant increase in the injected hormone confirming the efficacy of this method; however, this did not result in significantly elevated scale concentrations of the same hormone in all cases. As the stress and reproductive axes are closely integrated, these findings are likely a result of interactions along the steroidogenic pathway indicating that a further investigation of the relationship between scale concentrations of these hormones and actual physiological processes is required. Nevertheless, the successful quantification of both stress and gonadal steroid hormones within the scale suggests that such measurements could provide a novel and informative tool in the assessment of long-term stress and the resulting effects on reproductive endocrinology in teleost fishes.

**Keywords:** biomonitoring; reproduction; scales; steroid hormone; stress

## 1. Introduction

As human populations continue to expand, the frequency and duration of stressors applied to fishes in both wild and captive settings is on the rise. The stress response in teleost fishes is mediated largely by the hypothalamic-pituitary-inter-renal (HPI) axis. This begins with the secretion of corticotropin releasing hormone (CRH) from the hypothalamus which acts on the pituitary to trigger the release of adrenocorticotropic hormone (ACTH) [1]. ACTH then stimulates the steroidogenic cells of the inter-renal tissue resulting in the production of cortisol [2,3]. Cortisol is a glucocorticoid steroid hormone with many roles in the stress response, one of which serves to equip the fish with sufficient energy to overcome the perceived stressor and eventually resume normal functioning [1]. However, if the duration of the stressor is prolonged and homeostasis is not restored, the state of stress becomes chronic and this often results in adverse effects to an organism's health [4,5].

While chronic stress has been shown to interfere with many physiological processes [6–9], perhaps most concerning are reproductive alterations due to their potential impact on both present and future populations [10,11]. As many reproductive events are energetically

costly, the dwindling of energy resources to mitigate prolonged stressors has the potential to dampen reproductive activity as trade-offs between reproductive success and growth or survival are then required [10]. This competition between stress and reproduction is highly studied and involves multiple interactions between hormones produced by the HPI and hypothalamic-pituitary-gonadal (HPG) axes. To quantify this competition, comparisons between glucocorticoid and androgen concentrations are often employed [12]. For example, negative correlations between cortisol and both testosterone and 11-ketotestosterone (11KT) concentrations have been observed in fishes [1,13,14]. 11KT is a steroid hormone that has a higher affinity for the androgen receptor than testosterone in teleost fishes and is thus considered their primary androgen. Teleost fishes also produce two unique progestogens often referred to as the maturation inducing steroids (MIS): 17,20β-dihydroxy-4-pregnen-3-one (DHP) and 17,20β,21- trihydroxy-4-pregnen-3-one (20βS) [15]. These hormones are produced from the highly abundant progestogen, 17-α-OH progesterone and play an important role in spermatocyte and oocyte maturation, the production of seminal fluid, and ovulation [16–18]. Both the MIS and tightly regulated concentrations of cortisol are required for proper egg maturation providing additional opportunity for stress to interfere with reproduction. While relationships between stressors and the progestogens are somewhat less clear than with androgens, stress induced increases in cortisol and 17-α-OH progesterone were reported in zebrafish (*Danio rerio*) in response to a stressor [19]. Relationships between cortisol and estrogen are equally studied and like cortisol-androgen relationships, increases in cortisol have been shown to decrease circulating estradiol and its receptor [20]. These impacts of chronic stress can then lead to more concerning effects such as reduced vitellogenin production in females [21], reduced gamete quality in both sexes [21,22] and even adverse outcomes to progeny [23–27].

Recent success with the use of fish scales as a medium for steroid hormone quantification in long-term stress assessment has been reported by several groups [28–38]. This includes cortisol and now more recently cortisone and DHEA, as was reported in our recent studies in goldfish (*Carassius auratus*) and rainbow trout (*Oncorhynchus mykiss*) [38]. To further expand this area of research, the following study sought to incorporate both stress and reproductive related steroids into long-term stress assessments using the fish scale. While the exact mechanism of hormone deposition within the scale has yet to be determined, this likely occurs via passive diffusion from the vascularized epidermis to the scale [38–40]. As such, we artificially elevated circulating cortisol, progesterone and testosterone in adult rainbow trout using intraperitoneally injected hormone dissolved in coconut oil to further explore this relationship as well as the relationships between stress and reproductive scale steroids.

## 2. Materials and Methods

### 2.1. Preliminary Hormone Assays

Rainbow trout have relatively small scales both in surface area and mass and thus provide approximately 200 mg of dry scale per fish. This limits the number of hormones that can be quantified per fish. As our previous studies have indicated that scale glucocorticoid analysis generally requires 50 mg of powdered scale per hormone [38,39], we performed a preliminary analysis of five gonadal steroids to determine which hormones would be used in the present study. Table 1 outlines the mass of powdered scale required for the analysis of estradiol, 17α-OH progesterone, progesterone, testosterone and 11KT. As estradiol and 17α-OH progesterone required more than 100 mg of powdered scale for reliable analysis we chose to focus our study on cortisol, progesterone, testosterone and 11KT to best use the limited scale mass obtained from each fish.

### 2.2. Treatment Groups

Adult rainbow trout (*n* = 60) with an average mass of 306.5 +/− 76.1 g were divided into six treatment groups. The control group received no treatment, and all other groups received a weekly intraperitoneal injection of 5 µL of coconut oil per g of body mass

with or without dissolved hormone for three weeks. The vehicle control group received untreated coconut oil, the cortisol-injected group received coconut oil with 8 mg mL$^{-1}$ of hydrocortisone, the progesterone and testosterone-injected groups received coconut oil with 4 mg mL$^{-1}$ of either hormone and finally, the mix-injected group received coconut oil with all three hormones at the previously mentioned concentrations. This resulted in a weekly dosage of 40 mg kg$^{-1}$ of cortisol and/or 20 mg kg$^{-1}$ of progesterone and testosterone. These dosages were chosen based on a previous study [41].

**Table 1.** Mass of dry scale collected from rainbow trout required for reliable quantification of hormone via ELISA.

| Hormone | Dry Mass Required (mg) |
|---|---|
| Estradiol | 100+ |
| 17-α-OH progesterone | 100+ |
| Progesterone | 50–100 |
| Testosterone | 50–100 |
| 11-Ketotestosterone | 50–100 |

Each treatment group was held in one half of a 700 L rectangular tank at 12 ± 1 °C with a 14:10 light:dark photoperiod. Fish were fed each morning at 8 am and monitored to ensure food intake remained constant throughout the experiment. Ammonia, pH, chlorine, nitrite, and nitrate were monitored at least once per week to ensure ammonia and chlorine levels were less than 0.1 mg L$^{-1}$, that pH remained within 6.5–8.4 and that nitrate and nitrite levels remained less than 0.3 mg L$^{-1}$. A 25% water change was also performed each day.

### 2.3. Serum Collection

Fish were sampled for blood on day 22 of the experiment, 7 days after the final coconut oil injection. Prior to blood collection trout were anesthetized using a solution of 100 mg L$^{-1}$ of MS-222. A sample of blood was then collected from the caudal artery using a syringe, ejected into a 5 mL plastic tube and then left to clot for 3 h at 4 °C. Next, blood samples were centrifuged to allow the collection of serum which was transferred to a 1.5 mL plastic tube and stored at −20 °C until further analyses. Blood sampling took approximately 30 s per fish and all blood sampling was completed within 2 h.

### 2.4. Scale Collection

Cervical severance was used to euthanize anesthetized fish following blood collection and prior to scale collection. Trout were then wiped down to remove excess mucus and the entire body of scales was collected by scraping the length of the fish's body towards the head with a metal spatula. Scales were then transferred to 5 mL plastic tubes and stored at −20 °C for a maximum of 5 days.

### 2.5. Scale Hormone Extraction and Quantitation

Scales from individual fish were analyzed for four hormones: cortisol, progesterone, testosterone and 11KT. Although fish were not directly treated with 11-ketotestosterone, this potent androgen in fishes arises from testosterone and was thus included to examine the relationship between these two hormones as well as the other injected steroids. Prior to hormone extraction, fish scales were washed and ground as described by Kennedy and Janz (2022) [38]. In brief, scales samples of approximately 200 mg were washed for 2.5 min three times with methanol. After each wash, methanol was decanted, scales were blotted dry and any visible debris (skin, etc.) was removed with forceps. Wash tubes were also rinsed with methanol between each wash and a fresh aliquot of methanol was used for each successive wash. Washed and dried scales were then ground using a Retsch ball mixer mill MM 400 until a fine powder was achieved.

To extract the desired hormone subsamples of 50 mg of powdered scale were transferred to a 1.5 mL microcentrifuge tube. The extraction process was the same for each hormone beginning with the addition of 1 mL of HPLC grade methanol/50 mg of sample, then vortexing briefly for 10–15 s. Samples were then placed in a rotator and left for 18 h to extract. Following this, samples were centrifuged for 15 min, and the supernatant was collected and transferred into glass culture tubes. Next, the extracts were dried at 38 °C under a gentle stream of nitrogen. One ml of HPLC grade methanol was then added back to the powdered samples followed by a 40 s vortex, the tubes were then centrifuged, collected, and dried as above. These steps were repeated twice for a total of three collections. To concentrate the desired analyte at the bottom of the tube, the sides were rinsed four times with successively lower volumes of HPLC grade methanol. Between each rinse, extracts were dried at 38 °C under a gentle stream of nitrogen gas. Extracts were then reconstituted in 200–350 μL of extraction buffer supplied by their respective EIA kits: Cortisol EIA kit (Oxford Biomedical Rochester Hills, MI, USA), Progesterone ELISA kit (Enzo Life Science, Farmingdale, NY, USA), Testosterone ELISA kit (Enzo Life Science, Farmingdale, NY, USA), and 11-Ketotestosterone ELISA kit (Cayman Chemical, Ann Arbor, MI, USA). Next, the sample tubes were gently vortexed and incubated for 12 h at 4 °C. After 12 h, they were removed from the refrigerator, vortexed again, and the entire sample was transferred into a 0.6 mL microcentrifuge tube. Finally, the samples were centrifuged for five minutes to remove any remaining powdered scale. The supernatant was then collected and transferred to a clean 0.6 mL tube. Samples were run in triplicate following the kit protocols in a Molecular Devices Spectra Max 190 microplate spectrophotometer.

### 2.6. Serum Hormone Extraction and Quantitation

Serum collected from individual fish was also analyzed for four hormones: cortisol, progesterone, testosterone and 11KT. A subsample of 100 μL of serum for each hormone was transferred to a glass test tube. One ml of diethyl ether was then added to the tube followed by a 40 s vortex. To allow the ether and aqueous phase to separate, the tubes were then left to stand for five minutes. The ether layer was collected by flash freezing the tubes in liquid nitrogen for 7–10 s to allow the upper ether layer to be poured into a borosilicate glass test tube. The ether was then evaporated under a gentle stream of nitrogen gas at 50 °C. Once the aqueous phase thawed, the above steps were repeated for a total of three collections. The sides of the glass tube were then rinsed three times with decreasing volumes of ether (1 mL > 0.4 mL > 0.2 mL), drying the tube in between each rinse. Finally, the sample was reconstituted in 250–350 μL of EIA buffer from the kit to be used for analysis, vortexed gently for 40 s and incubated in the fridge overnight. The following day the sample was vortexed again for 40 s, transferred to a 0.6 mL plastic tube and stored at −20 °C until analysis using the respective ELISA kit.

### 2.7. Assay Validation

Intra- and inter-assay variation as well as parallelism were determined to ensure the accuracy, precision, and specificity of the hormone quantification methods. Extracts from multiple samples were pooled for intra-assay variation (*n* = 6) and inter-assay variation (*n* = 12), determined as the percent coefficient of variation (%CV, SD/mean). A concentrated sample was developed by collecting the extracts of multiple samples into a single glass tube to test parallelism by comparing the slopes of the standard curve and a serial dilution of the concentrated scale extract. Intra- and inter-assay variation for scale cortisol concentration was 7.9% and 11.2%, respectively, and 5.1% and 6.9% for serum cortisol, respectively. Intra- and inter-assay variation for scale progesterone concentration was 5.7% and 9.7%, respectively, and 6.4% and 14.7% for serum progesterone, respectively. For scale testosterone concentration, intra- and inter-assay variation was 9.4% and 10.5%, respectively, and 7.5% and 7.0% for serum testosterone, respectively. For scale 11KT concentration, intra- and inter-assay variation was 4.5% and 5.4%, respectively, and 6.0% and 9.6% for serum 11KT, respectively. Parallelism between extracted samples and the kit standard

curve was determined using a serial dilution of the pooled extract run in triplicate. The hill slope of both curves were then compared and if not significantly different, the curves were deemed parallel. Parallelism was observed between all standard curves and serially diluted extracts generated from both scale and serum samples. All validation steps were performed for all hormones. The limits of detection (LOD) for the cortisol, progesterone, testosterone and 11KT assays were 0.00510 ng mL$^{-1}$, 0.00104 ng mL$^{-1}$, 0.00602 ng mL$^{-1}$, and 0.000615 ng mL$^{-1}$, respectively. Any extract with a hormone concentration below the LOD was assigned the value of the LOD. This was the case for the testosterone concentration of three serum samples and three scale samples.

### 2.8. Statistical Analyses

Prior to any statistical testing, all data sets were tested for normality and homoscedasticity using the Shapiro–Wilk and Bartlett's tests, respectively. All groups failed these tests, and thus a Kruskal–Wallis test was employed. Multiple comparisons were performed between each hormone injection group and both the control and vehicle control only using a Dunnett's test. The control group was also compared to the vehicle control. Differences between groups were deemed significant at $p < 0.05$.

Correlations between serum and scale hormone concentrations were performed for all injected hormones (cortisol, progesterone, and testosterone) using the Spearman's rank correlation test. These correlations were deemed significant at $p < 0.05$.

## 3. Results

### 3.1. Serum Hormone Concentrations

#### 3.1.1. Cortisol

Serum cortisol concentrations were significantly elevated in the vehicle control, cortisol-injected, progesterone injected and mix-injected groups when compared to the control group (Figure 1A, $p < 0.05$). No other comparisons were statistically significant (Figure 1A, $p > 0.05$).

#### 3.1.2. Progesterone

Serum progesterone concentrations were significantly elevated in the progesterone and mix-injected groups when compared to both the control and vehicle control groups (Figure 1B, $p < 0.05$). No other comparisons were statistically significant (Figure 1B, $p > 0.05$).

#### 3.1.3. Testosterone

Serum testosterone concentration was significantly elevated in the testosterone-injected group when compared to both the control and vehicle control groups (Figure 1C, $p < 0.05$). No other comparisons were statistically significant (Figure 1C, $p > 0.05$).

#### 3.1.4. 11KT

Serum 11KT was significantly elevated in the testosterone and mix-injected group when compared to both control and vehicle control groups (Figure 1D, $p < 0.05$). No other comparisons were statistically significant (Figure 1D, $p > 0.05$).

### 3.2. Scale Hormone Concentrations

#### 3.2.1. Cortisol

Scale cortisol concentration in the testosterone-injected group was significantly lower than the vehicle control (Figure 2A, $p < 0.05$). Notably, the $p$-value obtained when comparing the scale cortisol concentration in the vehicle control to that of the progesterone-injected group was near significant ($p = 0.0576$). No other comparisons were statistically significant (Figure 2A, $p > 0.05$).

### 3.2.2. Progesterone

Scale progesterone concentration was significantly elevated in the mix-injected group when compared to the control group (Figure 2B, *p* < 0.05). No other comparisons were statistically significant (Figure 2B, *p* > 0.05).

### 3.2.3. Testosterone

Scale testosterone concentrations were significantly lower in the cortisol and progesterone-injected groups when compared to both the control and vehicle control groups (Figure 2C, *p* < 0.05). No other comparisons were statistically significant (Figure 2C, *p* > 0.05).

### 3.2.4. 11KT

Scale 11KT concentrations were significantly elevated in the progesterone, testosterone, and mix-injected groups when compared to the control group (Figure 2D, *p* < 0.05). Scale 11KT was also significantly elevated in the testosterone and mix-injected groups when compared to the vehicle control group (Figure 2D, *p* < 0.05). The *p*-value generated when comparing the progesterone-injected group to the vehicle control group also verged on significance (*p* = 0.0711). No other comparisons were statistically significant (Figure 2D, *p* > 0.05).

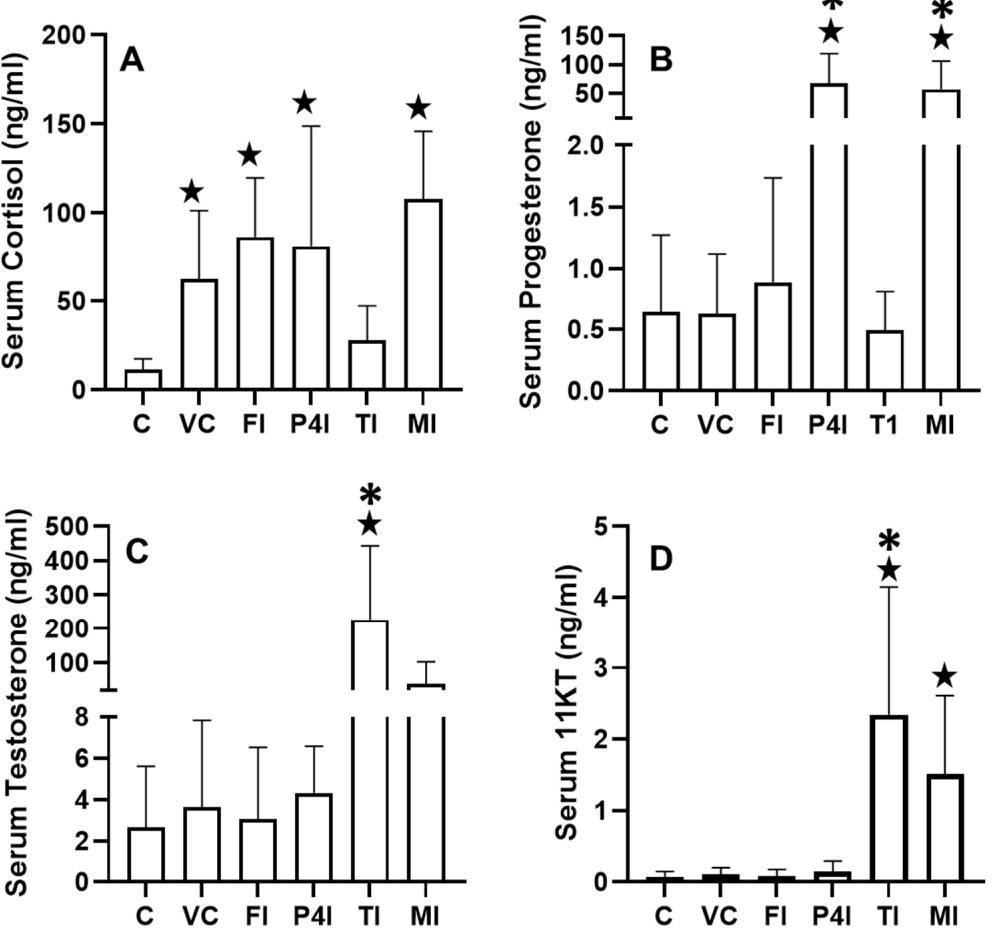

**Figure 1.** Serum hormone concentrations in control (C), vehicle control (VC), cortisol-injected (FI), progesterone injected (P4I), testosterone-injected (TI), and mix-injected (MI) fish: (**A**) cortisol, (**B**) progesterone, (**C**) testosterone, and (**D**) 11KT concentrations presented as the mean with error bars representing the standard deviation. All injected groups were compared to control and vehicle control only. Stars indicate significant difference from control, and asterisks indicate significant difference from vehicle control (*p* < 0.05; *n* = 49 fish).

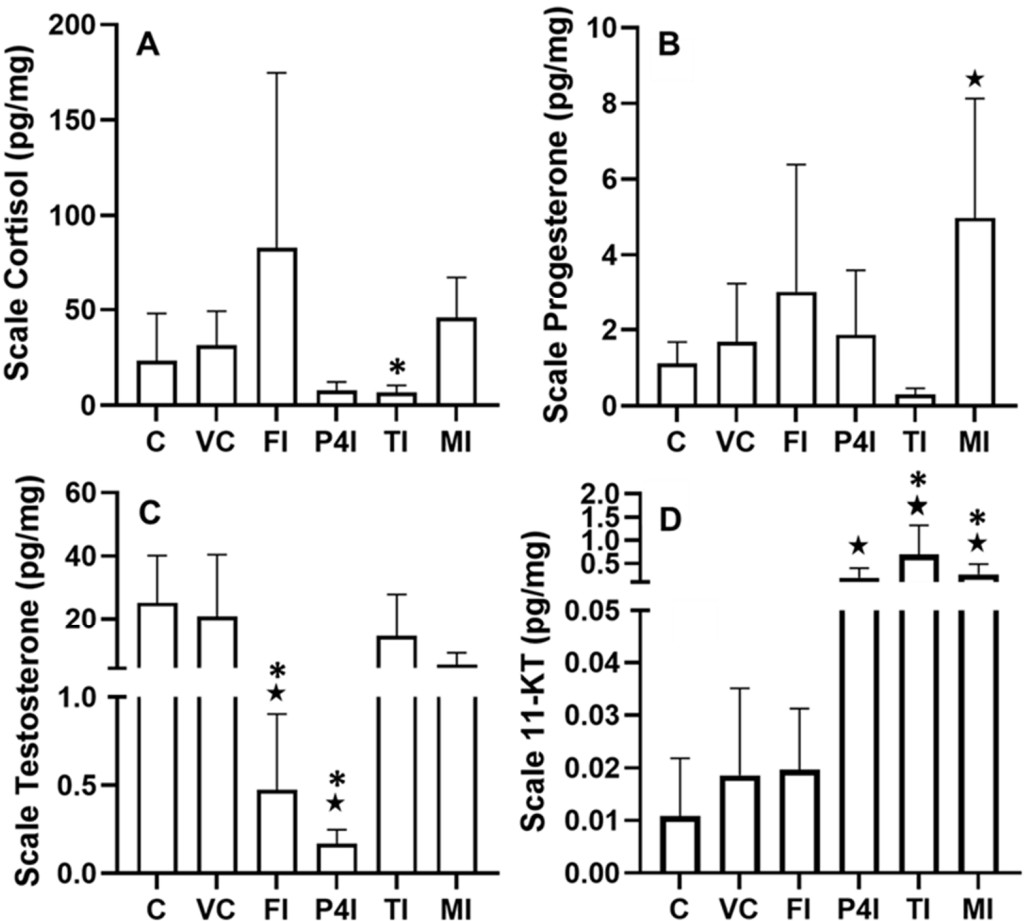

**Figure 2.** Scale hormone concentrations in control (C), vehicle control (VC), cortisol-injected (FI), progesterone-injected (P4I), testosterone-injected (TI), and mix-injected (MI) fish: (**A**) cortisol, (**B**) progesterone, (**C**) testosterone, and (**D**) 11KT concentrations presented as the mean with error bars representing the standard deviation. All injected groups were compared to the control and vehicle control only. Stars indicate significant difference from control, and asterisks indicate significant difference from vehicle control ($p < 0.05$; $n = 51$ fish).

### 3.3. Serum-Scale Correlations

#### 3.3.1. Cortisol

The correlation between serum and scale cortisol concentrations produced a spearman r coefficient ($r_s$) of 0.337 and was statistically significant (Figure 3A, $p < 0.05$).

#### 3.3.2. Progesterone

The correlation between serum and scale progesterone concentrations produced a spearman r coefficient ($r_s$) of 0.529 and was statistically significant (Figure 3B, $p < 0.05$).

#### 3.3.3. Testosterone

The correlation between serum and scale progesterone concentrations produced a spearman r coefficient ($r_s$) of 0.323; however, this correlation was not statistically significant (Figure 3C, $p > 0.05$).

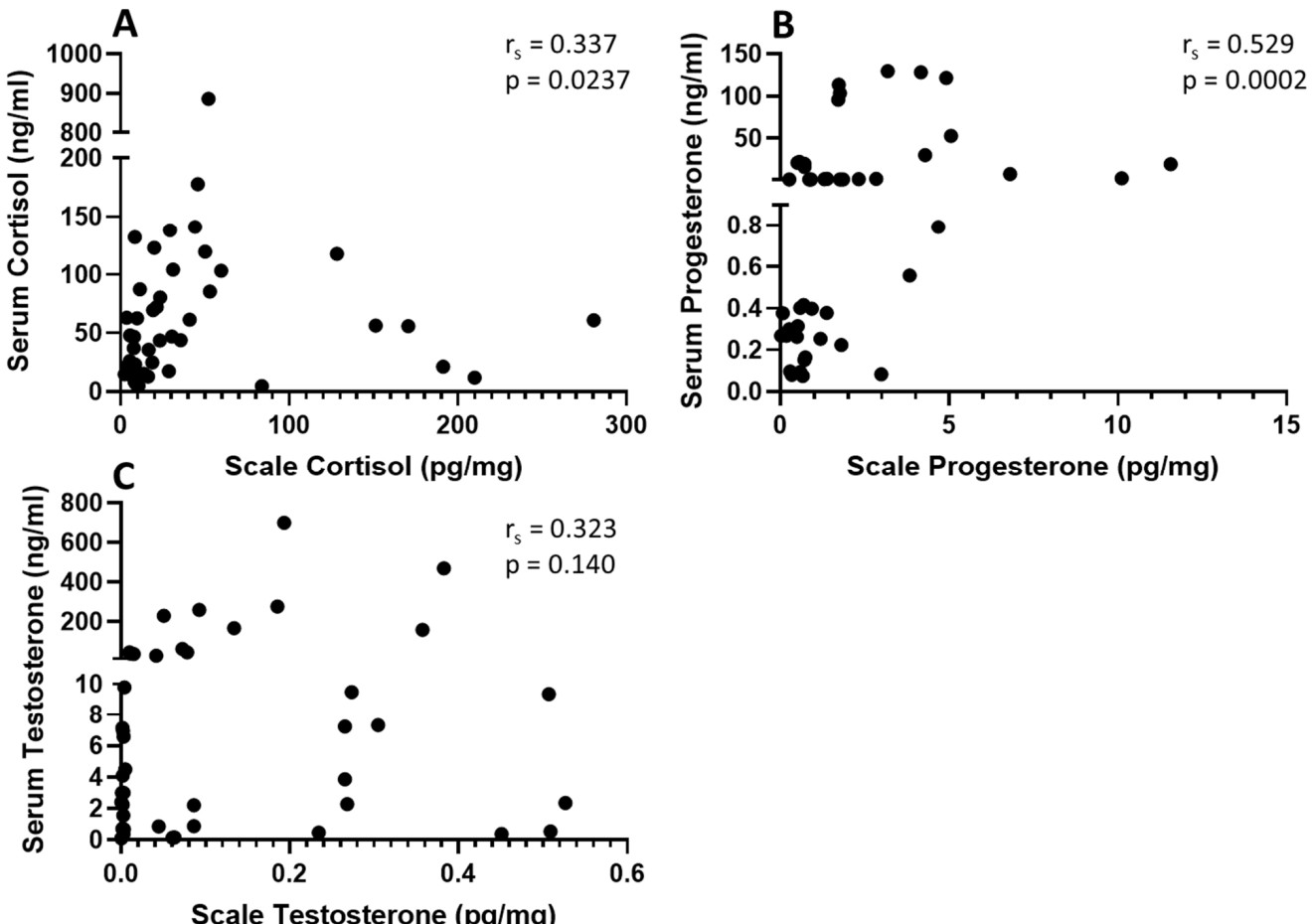

**Figure 3.** Correlations between serum and scale cortisol (**A**), progesterone (**B**), and testosterone (**C**) concentrations in all treatment groups. Significant correlations were observed between serum and scale cortisol and progesterone concentrations ($p < 0.05$) but not serum and scale testosterone concentrations ($p > 0.05$, $n = 44$–45 fish).

## 4. Discussion

In combination with our previous studies, we have now quantified eight different steroid hormones within the fish scale [38,39]. This confirms that like hair and feathers, scales incorporate and retain steroid hormones for a longer period than blood, allowing them to serve as a biomonitoring medium in a wide variety of species and contexts. While estradiol and 17-$\alpha$-OH progesterone were not used in this study due to the high scale mass required for their quantification, we recommend further investigation of these hormones in other studies and other species of fish as these hormones are particularly relevant to fish reproduction.

In all cases, single-hormone injections were successful in significantly elevating serum hormone concentrations in comparison to controls. However, in addition to being elevated in the cortisol and mix-injected groups when compared to the control group, serum cortisol was significantly elevated in the vehicle control and progesterone-injected group. As the vehicle control, the cortisol, progesterone, and mix-injected groups all had significantly elevated serum cortisol, it is likely that the injections acted as a stressor to the fish; however, the duration of this stressor is difficult to determine. The serum cortisol elevations were not reflected in scale samples suggesting that the serum cortisol elevations were a result of a relatively short period of stress induced by the accumulation of coconut oil within the intraperitoneal cavity near the end of the experiment. As evidenced by the growing body of knowledge pertaining to chronic stress and scale cortisol concentration, if the coconut oil injections had induced a state of chronic stress, we would have seen significant elevations

in scale cortisol as well [28–31,33,38,42]. This also supports the notion that scale cortisol concentration is relatively unaffected by acute stressors as demonstrated previously [42]. While serum cortisol concentration was significantly elevated in almost all groups this effect was not observed in the testosterone-injected group likely due to previously mentioned negative interactions often observed between cortisol and testosterone. An earlier study reported that testosterone-impregnated cocoa butter injections resulted in a reduced cortisol response after a one-hour confinement stress in rainbow trout [13]. This interaction was also apparent in scales as scale cortisol was significantly lower than vehicle controls in the testosterone-injected group suggesting that scale cortisol and testosterone relationships will play an important role in monitoring chronic stress and reproduction in fishes as it does in mammals [12].

Notably, although the change only bordered on significance, a suppression in scale cortisol was also produced by progesterone injections. While stress and reproduction are generally considered competitive processes, effects on scale cortisol produced by progesterone injections in this study are more difficult to explain. As progesterone acts as a precursor for most steroid hormones, an increase in scale cortisol upon progesterone injection could be expected [2]. Cortisol regulates its own production at all steps along the HPI axis and even at the level of 11β-HSD2, as reported by Alderman and Vijayan (2012) who discovered glucocorticoid response elements in the promotor of the 11β-HSD2 gene [43]. Increased transcription of the 11β-HSD2 gene and thus the conversion of cortisol to cortisone could therefore be the cause of our observed decreases in scale cortisol in the progesterone-injected group. Other possible means of 11β-HSD2 increases could come from the relationship between cortisol and the MIS. As previously mentioned, progesterone can be converted to one of two MIS: DHP or 20βS via the highly abundant intermediate 17-α-OH progesterone [15]. In rainbow trout, cortisol and DHP have both been shown to be involved in oogenesis [20,44,45]. As cortisol's effects on reproductive processes such as oogenesis can become negative, concentrations of cortisol are highly regulated during this period [20]. Upregulation of 11β-HSD2 during reproduction has thus been proposed as a means of protecting the ovaries from the damaging effects of increased cortisol. This suppression of scale cortisol induced by progesterone injection observed here could therefore be a result of 11β-HSD2 upregulation induced by progestogens.

As was intended, progesterone injections were successful in elevating serum progesterone in both the progesterone-injected and mix-injected groups when compared to the controls. However, this was not wholly reflected in scales, as scale progesterone was only significantly elevated in the mix-injected group. As outlined above, this could be a result of progesterone's conversion to other steroid hormones such as cortisol or DHP prior to incorporation into the scale. Notably, while progesterone injections resulted in a decrease in scale cortisol, cortisol injections produced the opposite effect and resulted in a three-fold increase in scale progesterone when compared to controls. While this increase was not statistically significant, stress-induced cortisol increases have been shown to result in increased 17-α-OH progesterone in zebrafish suggesting a possible positive correlation between cortisol and progestogens [19]. This could also contribute to the significantly elevated scale progesterone observed in the mix group as cortisol was injected alongside progesterone and testosterone in this treatment. Other notable changes in scale progesterone include the visible, although not significant, decrease in the testosterone-injected group. Specific inhibitory effects of testosterone on progesterone production are unlikely however, as all steroid hormones arise from cholesterol, an increase in any steroid hormone could provide negative feedback that results in a decrease in the conversion of cholesterol to pregnenolone and thus the production of other steroids [46].

Like the other injected groups, testosterone-injections were successful in elevating serum testosterone in the testosterone-injected group when compared to the controls. This however was not the case for the mix-injected group, likely due to the previously outlined negative interactions between cortisol and testosterone. By contrast, testosterone concentrations measured in the scale showed largely different results with scale testosterone

in both the testosterone and mix-injected groups not significantly elevated in comparison to controls. Testosterone acts as a precursor for 11KT, which is considered the primary androgen in teleost fishes. Therefore, these results are likely a consequence of testosterone conversion to 11KT. This is supported by both the serum and scale 11KT data which showed significant or near significant elevations in the testosterone and mix-injected groups when compared to controls. This could also relate to the previously discussed potential increase in 11β-HSD 2 triggered by DHP as 11KT formation from testosterone involves this same enzyme [2]. In addition, the progesterone-injected group showed significantly elevated scale 11KT, suggesting that progesterone is also acting as a precursor for 11KT. If the injected progesterone was converted to 11KT, this would also explain the near significant decrease in scale cortisol in the progesterone-injected group. In addition, this could aid in explaining the negative testosterone-progesterone interactions observed in both the testosterone and progesterone-injected groups, however this likely involves complex interactions and feedback loops along to steroidogenic pathway unable to be discerned by the data collected in this study.

As has been previously outlined, in order for these methods to be successful in diagnosing physiological change, a better understanding of blood-scale partitioning is required. In most cases, treatment groups that were injected with a particular hormone showed significantly elevated circulating concentrations of said hormone confirming that the hormone injections were successful. However, this did not produce consistent elevations in the scale concentrations of the same hormone. Regardless, significant correlations between serum and scale cortisol as well as progesterone concentrations were observed. While this was not the case for serum and scale testosterone correlations this can likely be explained by the observed conversion of testosterone to 11KT. Although progesterone was quantifiable within the scale, interactions between progesterone and the other steroid hormones were less clear. Relationships between circulating and scale progesterone were also less discernable as some progesterone appears to have been converted to 11KT prior to scale incorporation. However, these effects may have resulted from the artificial elevation of our injected hormones and thus these concepts need further exploration in natural settings.

## 5. Conclusions

This study introduced three new steroid hormones to be utilized in long-term stress assessments that incorporate fish scales as a sample media. Recent studies have shown that fish scale cortisol concentration can be used to diagnose chronic stress in fishes. The results presented herein suggest that the addition of scale reproductive steroids will provide an extended evaluation of chronic stress that includes alteration to reproductive endocrinology. Although these concepts require further investigation, their implementation as a tool for the biomonitoring of fishes has the potential to contribute to conservation efforts.

**Author Contributions:** Conceptualization, E.K.C.K. and D.M.J.; methodology, E.K.C.K. and D.M.J.; data curation, E.K.C.K.; writing—original draft preparation, E.K.C.K.; writing—review and editing, D.M.J.; supervision, D.M.J.; funding acquisition, D.M.J. All authors have read and agreed to the published version of the manuscript.

**Funding:** This research was funded by a Natural Sciences and Engineering Research Council of Canada (NSERC) Discovery Grant to D.M.J. (RGPIN-2016-05131).

**Institutional Review Board Statement:** This study was conducted under Animal Use Protocol 20200118 approved on 6 January 2021 by the Animal Research Ethics Board of the University of Saskatchewan.

**Informed Consent Statement:** Not applicable.

**Data Availability Statement:** Not applicable.

**Conflicts of Interest:** The authors declare no conflict of interest.

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
