# Peer review of "The Use of Fish Scale Hormone Concentrations in the Assessment of Long-Term Stress and Associated Adverse Effects on Reproductive Endocrinology"

_fishes, doi:10.3390/fishes7060393_

Round 1

Reviewer 1 Report

The present study evaluated the levels of cortisol and reproductive-related hormones in trout scales after artificial injection with the same hormones. This is regarding validating a method by which stress could be monitored with the non-invasive method.

The experiment is well designed and well written, meanwhile, the only obstacles is the few collected results only Table 3 (is the actual results). Accordingly, this paper could be accepted after minor correction as short communication not full-length article.

Minor comments

The fish scale or scales

The introduction is well written, but the last paragraph needs to support the physiological process by which any metabolites are incorporated in the scale body. Suggested Ref. “https://www.ncbi.nlm.nih.gov/pmc/articles/PMC6561483/”

L 119: “trout were anesthetized two at a time” revise this sentence.

L 129: “towards the tail with” or “towards the head with” .

L 283: “the duration of said stressor is difficult to determine” the use of “Said” is appropriate.

Author Response

Reviewer 1:

The experiment is well designed and well written, meanwhile, the only obstacles is the few collected results only Table 3 (is the actual results). Accordingly, this paper could be accepted after minor correction as short communication not full-length article.

Response: This manuscript now includes 3 figures (see reviewer 3 response) and presents data collected from six different treatment groups in two different sample media. We believe this is sufficient data for consideration as a full-length article.

The introduction is well written, but the last paragraph needs to support the physiological process by which any metabolites are incorporated in the scale body. Suggested Ref. https://www.ncbi.nlm.nih.gov/pmc/articles/PMC6561483/

Response: Additional information on the proposed mechanism of hormone transfer from blood to scale has been added along with the suggested reference (Lines 81-86).

L 119: “trout were anesthetized two at a time” revise this sentence.

Response: Done (Lines 124-125)

L 129: “towards the tail with” or “towards the head with” .

Response: This sentence was corrected, it now reads “towards the head with..” (Line 134).

L 283: “the duration of said stressor is difficult to determine” the use of “Said” is appropriate

Response: The wording of this sentence was edited for clarity (Line 310).

Reviewer 2 Report

This manuscript provides intersting results, as all the methods concerning the less-invasive methods for sampling or stress assessment in fish are interesting and deserves research and publication.

The work has been adequately performed both regarding the overall design and the methodology used, in particular the hormonal assessment. Nevertheless, results are not conclusive and it would be desired to deep in this type of work.

There are some details that are not clear to this reviewer. One is the fact that water quality was assessed only once a week. This would be coorect if quality was maintained at right levels, otherwise the timing is insufficient. Authors should provide the mean reference values obtained. A second one is that the authors mention that the blood sampling was performed in two hours, but they do not detail how lonh it took for one single fish. Since cortisol is secreted in about few minutes, the sampling may substantially affect the levels of hormones, rising the levels in sacles. This should be also considered in the discussion.

Author Response

Reviewer 2:

There are some details that are not clear to this reviewer. One is the fact that water quality was assessed only once a week. This would be coorect if quality was maintained at right levels, otherwise the timing is insufficient. Authors should provide the mean reference values obtained.

Response: Water quality was assessed a minimum once a week but the standard is three times. Ranges for the water quality parameters were also added to lines 117-120.

A second one is that the authors mention that the blood sampling was performed in two hours, but they do not detail how lonh it took for one single fish. Since cortisol is secreted in about few minutes, the sampling may substantially affect the levels of hormones, rising the levels in sacles. This should be also considered in the discussion.

Response: This additional info was added on lines 128-129 of the methods. There is also already a discussion regarding acute stressor cortisol transfer to scales in the discussion section of the manuscript (Laberge et al. 2019 demonstrated that acute stressors do not result in significant changes to scale cortisol so this is unlikely to be a problem).

Reviewer 3 Report

This research is novel because it is conducted from a new perspective of quantifying hormones present in scales. However, there is a need to show a correlation between hormone concentrations in the blood and those present in the scales. It should be shown whether or not the hormone concentration present in the scales is higher when the hormone concentration in the blood is higher.

Author Response

Reviewer 3:

This research is novel because it is conducted from a new perspective of quantifying hormones present in scales. However, there is a need to show a correlation between hormone concentrations in the blood and those present in the scales. It should be shown whether or not the hormone concentration present in the scales is higher when the hormone concentration in the blood is higher.

Response: We thank the reviewer for this suggestion. A third figure was added to depict serum-scale hormone correlations (Figure 3). Additional information on the tests use to assess such correlations was also added to the methods section (Lines 217-219) and a discussion of these results was added on Lines 394-397.
